# Comparative Study of Intestinal Microbiome in Patients with Ulcerative Colitis and Healthy Controls in Korea

**DOI:** 10.3390/microorganisms11112750

**Published:** 2023-11-11

**Authors:** Kyung-Hyo Do, Seung-Hyun Ko, Ki Bae Kim, Kwangwon Seo, Wan-Kyu Lee

**Affiliations:** 1College of Veterinary Medicine, Chungbuk National University, Cheongju 28644, Republic of Korea; pollic@chungbuk.ac.kr; 2GutBiomeTech Co., Ltd., Cheongju 28644, Republic of Korea; 3Department of Internal Medicine, Chungbuk National University Hospital, Cheongju 28644, Republic of Korea

**Keywords:** ulcerative colitis, intestinal microbiota, phylum, bacteria, Korean, bacterial abundance

## Abstract

Ulcerative colitis (UC) poses a contemporary medical challenge, with its exact cause still eluding researchers. This is due to various factors, such as the rising incidence, diagnostic complexities, and difficulties associated with its management. We compared the intestinal microbiome of patients with UC to that of healthy controls to determine the qualitative and quantitative changes associated with UC that occur in the intestinal microbiota. The intestinal bacterial abundance in 40 Korean patients with UC and 25 healthy controls was assayed using via next-generation sequencing. There were five major phyla in both groups: Firmicutes (UC patients: 51.12%; healthy controls: 46.90%), Bacteroidota (UC patients: 37.04%; healthy controls: 40.34%), Proteobacteria (UC patients: 6.01%; healthy controls: 11.05%), Actinobacteriota (UC patients: 5.71%; healthy controls: 1.56%), and Desulfobacteriota (UC patients: 0.13%; healthy controls: 0.14%). Firmicutes was more prevalent in patients with UC (51.12%) compared to that of healthy controls (46.90%). Otherwise, Bacteroidota was more prevalent in healthy controls (40.34%) compared to patients with UC (37.04%). Although there was no significant difference, our results showed a substantially lower gut microbiome diversity in patients with UC (mean: 16.5; 95% confidence interval (CI) = 14.956–18.044) than in healthy controls (mean: 17.84; 95% CI = 15.989–19.691), the beta diversity and the flora structure of the microbiome in patients with UC differed from those in healthy controls. This will be helpful for the development of new treatment options and lay the groundwork for future research on UC. To understand the disease mechanism, it is essential to define the different types of microbes in the guts of patients with UC.

## 1. Introduction

Ulcerative colitis (UC) is a persistent inflammatory bowel disease (IBD) affecting the colon, characterized by ongoing superficial inflammation of the mucous membrane [1,2]. Characteristic symptoms of UC include bloody diarrhea, fecal urgency, and tenesmus [3]. The duration of this illness is lengthy and does not resolve spontaneously. Individuals suffering from UC frequently experience recurrent episodes, necessitating lifelong treatment that significantly diminishes their quality of life. As a result, the World Health Organization has classified UC as a contemporary challenging medical condition [3]. The frequency of UC has been rising since the mid-twentieth century in Western societies, such as those found in North America, Europe, Australia, and New Zealand [4]. The prevalence of UC has steadily increased over the past decade, affecting millions of people worldwide [4,5]. Certain scientists assumed that UC represents a multifaceted systemic disorder intricately linked to various contributory factors, including disturbances in gut microbiota stability, immune system imbalances, genetic predisposition, and unhealthy lifestyle choices [6,7,8,9]. Nonetheless, the exact causes and mechanisms underlying this condition remain incompletely understood as of the present time [10,11,12,13]. Hence, there is a requirement for additional foundational research endeavors to address the above-mentioned queries.

As access to next-generation sequencing (NGS) increases, metagenomics research on integral microflora becomes more active. Utilizing NGS techniques to identify bacteria through their 16S rRNA gene sequences surpasses the constraints of culture-based methods. This approach represents an advanced tool for the culture-independent analysis of intricate microbial communities. Recent evidence has emerged regarding the influence of microbiota on the development of this disease [3,14]. The Human Microbiome Project established the first standards of bacterial Phylum abundance in healthy individuals [15]. The human intestinal microbiota in healthy individuals is typically composed of approximately 400 to 500 different bacterial species, with the majority belonging to four main phyla: Firmicutes, Bacteroidetes, Actinobacteria, and Proteobacteria [16]. However, subsequent studies on healthy populations have confirmed these findings, showing that Firmicutes is the dominant bacterial phylum and, when combined with Bacteroidetes, constitutes nearly 90% of the intestinal microbiota [3,16,17]. Several researchers have investigated the intestinal microbiota to clarify the pathogenesis of UC. Recent research reported that dysbacteriosis, or disturbed microbiome, is one of the most important environmental factors in UC, suggesting the probable involvement of the microbiota in the pathogenesis of UC [18,19]. Gryaznova et al. reported that there is a reduction in the number of specific Firmicutes bacteria, as well as a decrease in the functional diversity and stability of the intestinal microbial population, resulting in an increase in the number of *Bacteroidetes* bacteria and facultative anaerobes in individuals with UC [3]. Hörmannsperger et al. reported that some immunodeficient mice kept under conventional conditions developed chronic colitis spontaneously; however, if mice were kept under germ-free conditions or treated with long-term antibiotic therapy, the spontaneous development of colitis would be prevented [19]. Numerous investigations have demonstrated that bacterial strains obtained from mice with UC can induce intestinal inflammation in animals with immune deficits, even in wild mice [19]. The existing information from animal studies has highlighted the need for a full understanding of the composition and modifications of the intestinal microbiota in patients with UC to identify the likely causes and develop potential therapeutics [20].

Zakerska-Banaszak et al. reported that patients with UC showed alterations in the richness and composition of the intestinal microbiota at different taxonomic levels compared to non-UC individuals; a higher abundance of Proteobacteria, Actinobacteria, and Candidate, and lower abundance of Bacteroidetes and Verrucomicrobia were found in patients with UC [16]. Numerous studies have focused on the differences in bacterial diversity and composition. Sepehri et al. and Imhann et al. reported that there was a trend toward lower microbial diversity in patients with UC than in non-UC controls [21,22]. Furthermore, numerous studies conducted by other researchers have indicated that dysbiosis associated with UC often involves a decrease in beneficial commensal bacteria, particularly those within the Firmicutes and Bacteroides phyla, coupled with an increase in pathogenic species from the *Enterobacteriaceae* family [23,24,25,26]. C. Casen et al. reported 80% of IBD patients had dysbiosis characterized by a reduced abundance of Firmicutes [23] and Alan W Walker et al. reported that microbial diversity was reduced in IBD patients, and Firmicutes were reduced [24]. Zhang, M. et al. [25] and Zhang, S.-L. et al. [26] demonstrated that the gut microbiota of patients with UC had significant changes in amount of Bacteroides and had roles in the onset and progression of IBD.

However, contradictory results exist between studies, necessitating further studies on the gut flora of patients with UC. In a recent systematic review, significant heterogeneity was observed among studies on gut microbiota, and no consistent characteristics of irritable bowel syndrome-related gut microbiota have been identified [27]. Willing et al. reported that significant differences between individuals with UC and healthy individuals were not observed and assumed that this lack of distinction was attributed to the fact that environmental exposure during early childhood exerted a more profound and enduring influence on the gut microbiota compared to disease status [28]. Also, He, X.X. et al. reported that the abundance of Firmicutes significantly increased in the remission stage of UC patients [20]. Therefore, further research is required to determine how the microbiome differs in the gastrointestinal tract of patients with UC. This will be helpful for the development of new treatment options and lay the groundwork for future research on UC. The purpose of this study was to compare the intestinal microbiome of patients with UC to that of people without UC; specifically, we aimed to determine the qualitative and quantitative changes associated with UC in the intestinal microbiota at different taxonomic levels. To understand the disease mechanism, it is essential to define the different types of microbes in the guts of patients with UC.

## 2. Materials and Methods

### 2.1. Ethics Statement

This study was approved by the Institutional Review Board of Chungbuk National University (Registration number: CBNUH CTC-21-04; Effective date: 11 November 2021).

### 2.2. Patients and Healthy Controls

We recruited 40 patients with UC and 25 healthy controls who were not taking any medicine nor receiving any treatments. A total of 40 Korean patients (19 female and 21 male) with ulcerative colitis (UC) under the care of Chungbuk National University Hospital and 25 healthy controls (17 female and 8 male) were enrolled in this study. Every participant provided written informed consent. The participants were recruited between November 2021 and June 2023. Blood samples were collected from the 40 patients with UC and 25 healthy controls to assay for complete blood cell counts and to perform biochemical analyses.

The diagnosis of UC relied on clinical, endoscopic, and histological information. The detailed characteristics of all patients with UC, including age, weight, height, body mass index, complete blood cell count, and blood chemical test results, are presented in Table 1. The exclusion criterion was antibiotic intake in the last month.

The healthy control group comprised 25 people (17 female and 8 male) from the Korean population who are not taking any medicine or receiving any treatments. The exclusion criteria were immunosuppressant or corticosteroid intake during the last six months. Fecal samples were taken from each participant and stored until analysis at −80 °C.

Before the experiment, we confirmed that there were no statistically significant differences in age, sex rates, and body mass index between the patients with UC group and healthy control group.

### 2.3. DNA Extraction

Fresh fecal samples from participants were collected in a sterile container, and then stored in the deep-freezer at −80 °C. Bacterial DNA was extracted from 200 mg of fecal sample collected from each individual using a Maxwell RSC Instrument (Promega, Madison, WI, USA) and a Maxwell RSC Fecal Microbiome DNA Kit AS1700 (Promega), according to the manufacturer’s protocol. After the extraction, the quality of the DNA was assayed using a Qubit dsDNA HS assay kit (Life Technologies, Gent, Belgium) in a Qubit fluorometer (Thermo Fisher Scientific, Waltham, MA, USA) according to the manufacturer’s protocol.

### 2.4. 16S rRNA Gene Sequencing

After quantification using a Qubit dsDNA HS assay kit (Life Technologies, Gent, Belgium) in a Qubit fluorometer (Thermo Fisher Scientific, USA), 65 16S V3–V4 amplicon libraries (40 libraries for patients with UC and 25 libraries for healthy controls) were prepared according to the Illumina metagenomic sequencing library construction workflow (Part # 15,044,223 Rev. B, Illumina, San Diego, CA, USA). Briefly, sequences of the V3–V4 region of the 16S rRNA bacterial gene were amplified using a KAPA HiFi HotStart ReadyMix PCR Kit (KAPA Biosystems, Wilmington, MA, USA) and the following primers (16S rRNA gene-specific sequences are underlined): forward 5′-TCGTCGGCAGCGTCAGATGTGTATAAGAGACAGCCTACGGGNGGCWGCAG-3′ and reverse 5′-GTCTCGTGGGCTCGGAGATGTGTATAAGAGACAGGACTACHVGGGTATCTAATCC-3′. Subsequently, a second PCR was carried out to attach the index adapters for sample multiplexing using a Nextera XT Index Kit (Illumina, San Diego, CA, USA). The concentration of each DNA library was quantified using a Qubit fluorometer (Thermo Fisher Scientific, USA). The DNA libraries were pooled at a concentrations 4 nM and denatured with 0.2 N NaOH. The final concentration of the library was 7 pM, the phiX control library was added at 30% (*v*/*v*) of the final loading samples as the same concentration (7 pM) of the library. The sequencing of the libraries was carried out on the MiSeq platform according to the manufacturer’s instructions, using a MiSeq Reagent Kit v3 (600 cycles) (Illumina).

### 2.5. Bioinformatic Analysis and Statistical Analysis

The 16S rRNA gene sequences were acquired and processed through the classify-sklearn naïve Bayes classifier workflow of QIIME 2 2023.7. The reads were de-multiplexed and trimmed. The forward and reverse reads were merged and matched to respective samples using their assigned indices. Subsequently, raw reads were trimmed using QIIME. Quality filtering was applied to the combined sequences, discarding sequences that did not meet the following criteria: sequence length <20 nucleotides or >300 nucleotides. Chimeric sequences were eliminated, and operational taxonomic units (OTUs) were clustered against the SILVA v138 99% full-length database. Prior to computing alpha and beta diversity statistics, the sequences were rarefied. The sufficiency of the sample size and estimation of species richness were determined using a rarefaction curve.

Differences in species richness and evenness scores, considering the sampling depth, were analyzed using the Chao 1 alpha diversity index. Alpha diversity statistics were calculated in QIIME 2. Beta diversity was assessed through the utilization of Bray–Curtis distances, and subsequently, a principal coordinate analysis was conducted. The differences in microbial diversity among samples (beta diversity) were measured using Bray–Curtis distances and represented visually through multidimensional scaling plots. Multifactorial permutational analysis of variance (PERMANOVA) was used to test the association between patients with and healthy controls using the QIIME 2 2023.7 (http://www.qiime2.org/, accessed on 22 September 2023). The intestinal bacterial abundance was analyzed using MicrobiomeAnalyst (https://www.microbiomeanalyst.ca/, accessed on 22 September 2023). Clustering analysis was used to distinguish between high- and low-abundance taxa, and color gradients were employed to indicate the degree of similarity in community composition among the various samples in the research. A cluster analysis was performed on the top 37 genera. Utilizing a Euclidean distance matrix, cluster analysis and dendrogram were conducted using the Ward method. The outcomes of the clustering were then merged with the relative abundance data of species at the genus levels within each sample.

## 3. Results

### 3.1. Clinical Data of Patients with Ulcerative Colitis and Healthy Controls

There was no significant difference in the female/male ratio (19/21 and 17/8, respectively), age (48.5 ± 17.3, and 43.8 ± 10.7, respectively), or body mass index (24.38 ± 3.81, and 23.30 ± 3.86, respectively) between patients with UC group and healthy control group.

The results of the complete blood cell count and biochemical analysis showed that there was no significant difference in the levels of white blood cell, red blood cell, hemoglobin, hematocrit, calcium, phosphorus, glucose, creatinine, estimated glomerular filtration rate (eGFR), blood urea nitrogen, uric acid, protein, aspartate aminotransferase (AST), alanine aminotransferase (ALT), alkaline phosphatase (ALP), or total bilirubin between patients with UC and healthy controls. Otherwise, the levels of platelet (patients with UC: 251.79 ± 56.39; healthy controls: 294.08 ± 56.84), cholesterol (patients with UC: 187.43 ± 37.77; healthy controls: 210.16 ± 49.21), and albumin (patients with UC: 4.48 ± 0.61; healthy controls: 4.81 ± 0.31) were significantly lower in patients with UC compared to those of the healthy control group.

### 3.2. Overall Composition of the Intestinal Microbiota in UC Patients and Healthy Controls

To visualize the distribution of the intestinal microbiota in the patients with UC and healthy controls at the phylum level and genus level, a stacked bar figure was prepared (Figure 1). As depicted in Figure 1, the primary phyla observed in both UC patients and healthy controls included Firmicutes (patients with UC: 51.12%; healthy controls: 46.90%), Bacteroidota (patients with UC: 37.04%; healthy controls: 40.34%), *Proteobacteria* (patients with UC: 6.01%; healthy controls: 11.05%), Actinobacteriota (patients with UC: 5.71%; healthy controls: 1.56%), and Desulfobacterota (patients with UC: 0.13%; healthy controls: 0.14%); however, their proportions varied. Specifically, the prevalence of Firmicutes exceeded that of Bacteroidota and Proteobacteria. The abundance of Proteobacteria was considerably higher in the healthy controls (patients with UC: 6.01%; healthy controls: 11.05%) and that of Actinobacteriota was considerably higher in the patients with UC (patients with UC: 5.71%; healthy controls: 1.56%); however, the differences were not statistically significant. At the genus level, although there were no significant differences, the abundance of *Bacteroides* (patients with UC: 33.35%; healthy controls: 37.26%), *Fecalibacterium* (patients with UC: 13.74%; healthy controls: 17.73%), and *Escherichia*_*Shigella* (patients with UC: 5.47%; healthy controls: 10.31%) were considerably higher in the healthy control group compared to the patients with UC group. The analysis of the samples showed large inter-individual variations in the composition of intestinal microflora at the phylum level among both the patients with UC and healthy controls, as depicted in Figure 2.

To visualize the distribution of the intestinal microbiota in patients with UC and healthy controls at the genus level, a heat map was prepared (Figure 3). Different individuals in both the patients with UC and in healthy control groups showed different abundance patterns in their intestinal microbiota. The cluster heatmap of the intestinal microflora between the patients with UC and healthy controls revealed that there are huge intra-individual differences regardless of UC status. The intestinal microbiota of the patients with UC and healthy controls revealed no clear differences. Also, there was no statistically significant phylogenetic group among the fecal samples from patients with UC and healthy controls.

### 3.3. Alpha Diversity and Beta Diversity

We investigated the community richness (alpha diversity) using the Chao 1 index of the patients with UC and healthy controls (Figure 4A). The Chao 1 index showed that the diversity of intestinal microbiota in the fecal samples was considerably reduced in the patients with UC compared to healthy controls; however, there were no significant differences (*p*-value: 0.30528). The 95% confidence interval of the Chao 1 index of the patients with UC was 14.955 to 18.044 (mean value: 16.5) while that of the healthy controls was 15.989 to 19.691 (mean value: 17.84).

To assess the degree of species overlap between the patients with UC and healthy controls, we calculated the beta diversity using the Bray–Curtis Index (Figure 4B). The X and Y axes represent two chosen spindles, with the percentages indicating the interpreted values for dissimilarities in sample composition. The scales on the X and Y axes are relative distances and lack practical significance. A shorter distance between two sample points signifies a higher similarity in their species composition. Through this, although there were no significant differences (Adonis R^2^ = 0.027771; F-value: 1.7995; *p*-value: 0.063), we found that the microflora structure of the patients with UC was different from that of the healthy controls.

## 4. Discussion

In this study, we investigated the microbiological composition of the intestines of patients with UC in Korea and compared the data with those of a healthy control group. To the best of our knowledge, this study represents the first analysis of the gut microbiota of Korean patients with UC and healthy controls, incorporating clinical data such as complete blood cell counts and biochemical analyses. The importance of the gut microbiome in human health and disease has been increasingly recognized in recent years. Studies have highlighted disturbances in the gut microbiological balance in inflammatory bowel disease, especially ulcerative colitis (UC), although the results are inconsistent [29]. In this study, although there were no significant differences, we found that the abundance at the phylum and genus levels, and the intestinal microflora structure of patients with UC were different from those of healthy controls, and community richness was lower in patients with UC. Continued research into the functional aspects of these microbial variations has the potential to reveal the complex mechanisms, offering new perspectives on managing UC and enhancing the overall health of those affected.

Age, gender, weight, height, and body mass index are risk factors for UC, and could affect the intestinal microflora [1]. To exclude the effect of these factors on intestinal microflora, we confirmed that there were no significant differences in age, gender, weight, height, and body mass index between the patients with UC and healthy controls.

In our current research, we observed a different pattern in the intestinal microbiota of patients with UC compared to healthy controls at both the phylum and genus levels (Figure 1). Khan et al. reported that the dysbiosis pattern most strongly associated with UC involved a reduction in the diversity of commensal bacteria, especially Firmicutes and Bacteroidota, along with a corresponding increase in bacterial species belonging to *Enterobacteriaceae* [29]. Eckburg et al. reported that the major bacterial phyla in intestinal microbiota are Firmicutes, Bacteroidetes, Actinobacteria, and Proteobacteria [30]. We also found that these phyla comprised more than 99% of the intestinal microbiota in both groups (patients with UC and healthy controls). Our study demonstrated that Firmicutes was the most prevalent phylum in both groups, and the patients with UC had a higher abundance of Firmicutes compared to healthy controls (patients with UC: 51.12%, and healthy controls: 46.90%, respectively). The levels of bile acids might be altered in patients with UC [1,8,20]. Changes in the composition of fecal bile acids might be involved in modulating inflammatory responses [6,8,20]. The primary bile acids are transformed into secondary bile acids by the intestinal microbiota through various reactions, such as deconjugation, dehydroxylation, esterification, and desulfatation [6,8,9]. Dehydroxylation is facilitated by the Firmicutes [3,8,9]. Therefore, an imbalance in gut microbiota could disrupt the processing of bile acids, potentially leading to elevated levels of secondary bile acids in patients with UC and contributing to the outbreak of UC.

Most studies conducted on UC have indicated a decrease in Bacteroidota in cases of UC [30,31,32,33,34]. Nomura et al. reported that Bacteroidetes species negatively correlated with UC [35]. Like other studies, we found that the second most abundant phylum, Bacteroidota (synonym Bacteroidetes), was considerably lower in the UC patients (37.04%) than in healthy controls (40.34%), although it was not statistically significant. Bacteroidetes, along with other beneficial bacteria, play a role in maintaining gut homeostasis by contributing to processes such as digestion, nutrient absorption, and immune regulation [36]. Changes in the composition of the gut microbiota, including a reduction in beneficial bacteria like Bacteroidetes, can disrupt these processes and potentially contribute to inflammation and other symptoms associated with ulcerative colitis [36]. Also, the phylum Bacteroidota is known to produce short-chain fatty acids, which can increase the number of colonic regulatory T cells by promoting the migration of extra-intestinal regulatory T cells [37]. Through this, we could assume that a lower abundance of Bacteroidota could have a negative impact on adaptive immune microbiota coadaptation and it could contribute to the outbreak of UC.

In this study, we found that the abundance of Proteobacteria was considerably lower (6.01%) in the UC patients compared to healthy controls (11.05%). There is still a lot of controversy over the abundance of Proteobacteria in patients with UC. Proteobacteria encompasses a variety of pathogenic strains that have the potential to play a role in gut inflammation. UC is characterized by persistent inflammation of the colon lining, and the presence of specific Proteobacteria species could potentially worsen this inflammatory process [32]. Some studies suggested that the presence and abundance of specific Proteobacteria can serve as biomarkers for UC [26,29,30,31,32]. Researchers have identified certain taxa within the phylum Proteobacteria that are more prevalent in patients with UC, and these markers can potentially be used for diagnostic or prognostic purposes [26,29,30,31,32]. However, Alam, M.T. et al. [38], Rehman, A. et al. [39], and de Meij, T.G.J. et al. [40] reported that patients with UC had a significantly lower abundance of Proteobacteria. Proteobacteria are commonly observed in higher quantities within the intestinal microflora of individuals who follow a high-sugar diet or are part of an obese population [41,42]. This suggests the importance of avoiding dietary factors that can disrupt the balance of Proteobacteria to prevent the development and progression of inflammatory conditions.

We also showed that the microbiota of patients with UC was more abundant in Actinobacteria (5.71%) relative to healthy controls (1.56%). This result aligns with the findings of a study conducted by American researchers involving 61 UC patients and 61 healthy individuals [32]. Actinobacteria are known to exert immunomodulatory effects within the gastrointestinal tract. In the context of UC, there is frequently an abnormal immune response to the gut microbiota, which plays a role in the inflammation associated with the disease [26,29,30,31,32]. Alterations in the Actinobacteria composition could potentially impact the local immune response, either intensifying or alleviating the inflammation [31].

There were interesting findings in the analysis at the genus levels between the patients with UC and healthy controls. Previous studies have reported a decreased number of members of the genus *Bifidobacterium* [38,43,44]. However, our results showed that the microbiota of patients with UC had a higher abundance of *Bifidobacterium* (5.28%) than that of healthy controls (1.9%). Although there are conflicting reports regarding the prevalence of *Bifidobacterium* [38,43,44], some of these data support previous findings indicating that patients with UC have higher levels of *Bifidobacterium* [45]. The genus *Bifidobacterium* is a beneficial probiotic that prevents intestinal inflammation by inducing intestinal IL-10-producing regulatory T cells and ameliorating colitis [45]. Patients with UC who participated in the study were treated with prebiotics and probiotics, which have a strong influence on the proliferation of *Bifidobacterium*. In addition, patients with UC frequently consume probiotics and prebiotics, even without a prescription [1,2,3]. This could be the reason for the increased *Bifidobacterium* [43,46].

Butyrate helps to stabilize the function of the intestinal barrier and decrease inflammatory processes. Patients with UC have been shown to have reduced levels of both butyrate-producing bacteria and butyrate content [2,47]. *Bacteroides* and *Faecalibacterium* are the key contributors to the production of butyrate and other short-chain fatty acids (SCFAs). These SCFAs serve a dual purpose, serving as an energy source for intestinal epithelial cells and exerting anti-inflammatory effects at the epithelial layer [47]. In this study, the patients with UC showed a considerably lower abundance of *Bacteroides* (33.36%) and *Faecalibacterium* (13.74%) compared to that of healthy controls (37.26%, and 17.73%, respectively). When butyrate levels are low in UC patients, several issues may arise. The survival and differentiation of colonic epithelial cells may be reduced, leading to an increased inflammatory response. Additionally, a gut microbiota imbalance may occur, with a possible overgrowth of harmful microorganisms. All of these factors can exacerbate inflammation in the colon, potentially leading to more severe UC symptoms [47].

Also, we found that patients with UC had a higher abundance of *Dialister* (4.49%) relative to healthy controls (0.62%). Based on information from existing literature, *Dialister* has the capacity to generate succinate and acetic acid [48]. The elevation in microbial production of these metabolites plays a role in the pathogenesis of UC, as they serve as pro-inflammatory signaling molecules [48]. Through this, increases in *Dialister* abundance among patients with UC suggest a potential link to the pathogenesis of the disease. Further investigations into the specific metabolic activities of *Dialister* in UC could provide valuable insights into the mechanisms underlying this association.

Alpha diversity represents the distribution of species abundance in a particular sample, based on species richness and evenness [16]. A recently published study of 132 patients with UC provided the most comprehensive description of host and microbial activities in UC, demonstrating that the gut microbiome is central to this disease [49]. We found that the microbiota of the patients with UC showed a lower alpha diversity based on the Chao 1 index compared to that of healthy controls (Figure 4A), indicating that the diversity of the intestinal microbiota was reduced in patients with UC [16,50], which is consistent with previous research on dysbiosis in UC [13,51]. This finding is consistent with the report by Michail, S. et al., which observed a significant reduction in the richness, evenness, and biological diversity of the intestinal microbiome in patients with UC when compared to healthy controls [3].

Also, we found that the different structure in intestinal microbiota between patients with UC and healthy controls, as shown by the principal coordinate analysis (beta diversity; Bray–Curtis Index; Figure 4B), indicating a correlation between different microbial phyla in both sample groups. Nevertheless, the dendrogram generated using the Ward clustering algorithm did not display distinct differences between the group of patients with UC and the group of healthy controls (Figure 3).

The diversity of intestinal microbiota has been suggested to be disrupted in patients with UC. Furthermore, additional research has demonstrated that the stability of the intestinal microbiota is decreased in individuals with UC [16,50]. The differences between patients with UC and healthy controls may result from changes in the gut microbiota caused by the disease [16]. This analysis demonstrated that certain correlations between individual units of the microbiota are not consistent and may be influenced by several factors. These factors may include but are not limited to the intestinal environment, the availability of specific endogenous substances, and the presence of the disease, all of which may strongly influence certain correlations between individual units of the microbiota [13].

In this study, differences in gut microbiota were observed in the abundance at the phylum and genus levels, as well as in the intestinal microflora structure between the patients with UC and healthy controls. Additionally, the patients with UC exhibited a lower community richness compared to healthy individuals. Our findings will provide a foundation for researching treatments focused on modifying the intestinal microbiome of patients with UC.

The limitations of this study include the small sample size and the employed methodology. In this study, we found that there was a difference in gut microbiota between patients with UC and healthy controls but statistical significance was not clearly demonstrated. In the future, using a larger sample size could provide us with greater statistical power. Also, because UC has many contributing factors including disturbances to gut microbiota stability, immune system imbalances, genetic predisposition, and unhealthy lifestyle choices, it is important to ensure that these factors are controlled in the sample. If microbial confirmation using culturing and identification had been performed, it would have been more helpful in understanding the gut microbiota of patients with UC and healthy controls. However, in this study, microbial confirmation was not conducted. In addition, the microbial community characterization was conducted using the MiSeq Illumina platform, based on the common V3–V4 fragments of the 16S rRNA gene. It has been shown that the degree of variation in different microbial taxa may be influenced by these regions (such as V1–V9) in the 16S rRNA gene [30]. Using primers that target specific groups of bacteria may lead to limitations in the detection and targeting of other microbial groups, thus potentially limiting the comparability of results between studies. In future research, it would be beneficial to investigate bacterial species, their functions, and their correlation with host genetic factors. This approach could enhance our understanding of the etiology of UC.

## 5. Conclusions

In conclusion, in this study, we aimed to compare the intestinal microbiota of patients with UC with that of individuals without UC. Although there were no significant differences in the abundance at the phylum and genus levels between patients with UC and healthy controls, the community richness of the intestinal microflora of patients with UC was less diverse and the flora structure of patients with UC was different from that of healthy controls. The gut microbiota plays a pivotal role in UC. Patients with UC often exhibit over-representation or under-representation of specific microbes within their gut microbiota composition. We assumed that dysbiosis in UC primarily involves a decrease in beneficial bacteria and an increase in harmful microbial groups. However, it is important to acknowledge that various factors, such as host genetic elements like the expression of specific inflammatory proteins, could influence the precise composition of gut microbiota. Thus, additional research is required to thoroughly understand the precise contribution of microbes in the etiology of UC. In the future, research focusing on bacterial species, their functions, and their relationship with host genetic factors will be essential in unraveling the underlying causes of UC. This approach holds great promise for advancing our understanding of the condition.

## Figures and Tables

**Figure 1 microorganisms-11-02750-f001:**
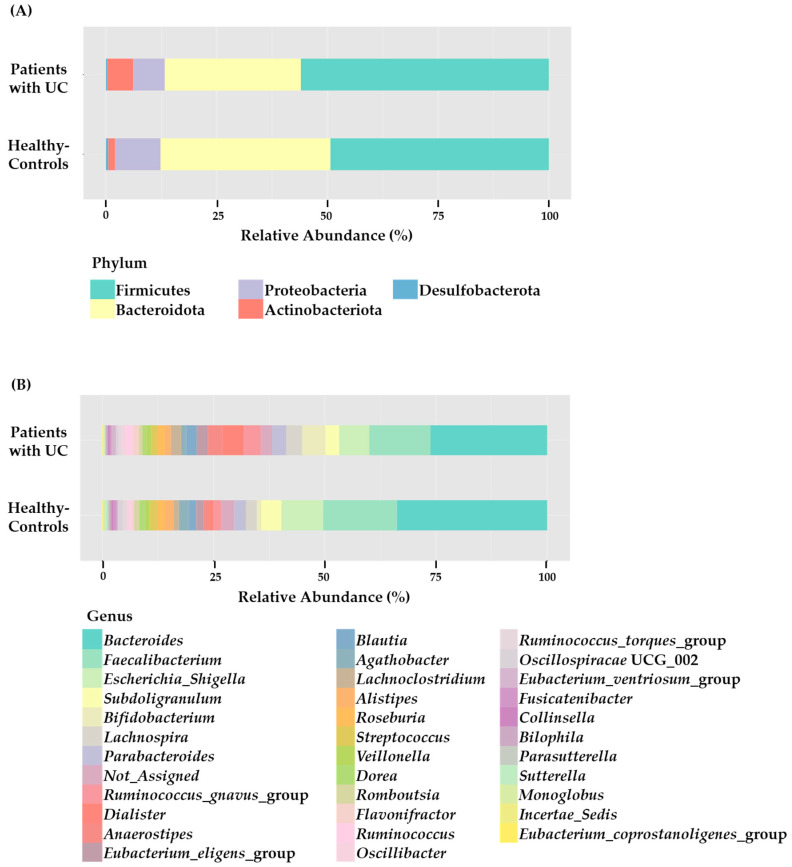
The microbiological compositions at the phylum (**A**) and genus (**B**) levels. The stacked bar shows the average relative abundances of all phyla and the most common genera identified in patients with ulcerative colitis (patients with UC) and healthy controls.

**Figure 2 microorganisms-11-02750-f002:**
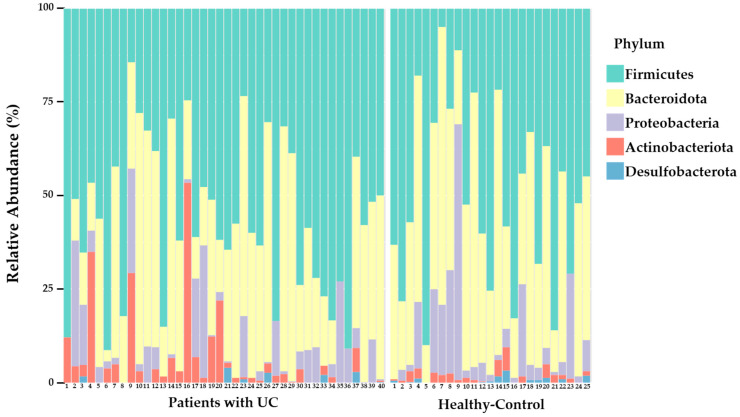
Individual profiles of gut microbial composition at the phylum level in the patients with ulcerative colitis (patients with UC) and healthy control group.

**Figure 3 microorganisms-11-02750-f003:**
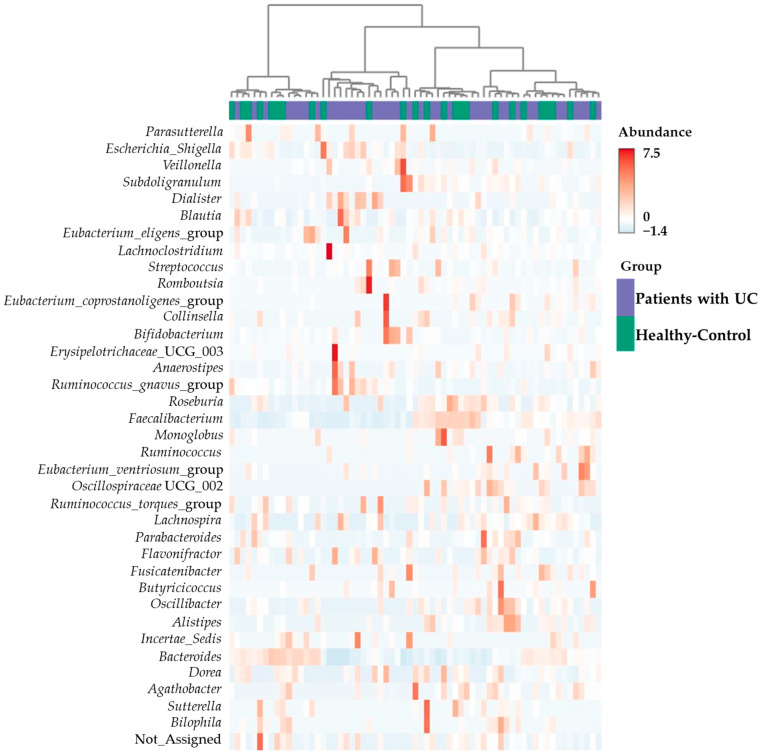
Cluster analysis and dendrogram of microbiota. Top: hierarchical clustering analysis of samples from patients with ulcerative colitis (patients with UC). Bottom: heatmap of the intestinal microbiota in fecal samples from patients with UC and healthy controls at the genus level.

**Figure 4 microorganisms-11-02750-f004:**
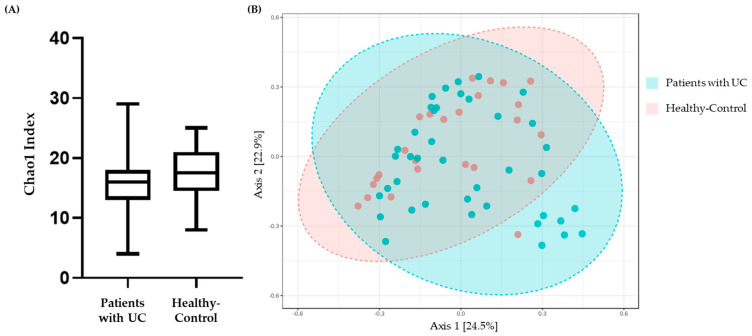
Alpha diversity (**A**) and beta diversity (**B**) analyses of the intestinal microbiota in fecal samples from patients with ulcerative colitis (patients with UC) and healthy controls. (**A**) Results of alpha diversity analysis of fecal samples using Chao 1 index. The *p*-value was calculated using the *t*-test. (**B**) Results of beta diversity analysis and principal coordinate analysis based on Bray–Curtis index. The *p*-value was calculated using PERMANOVA. Various sample groupings are indicated by points with distinct colors or shapes, and the horizontal and vertical axes represent relative distances.

**Table 1 microorganisms-11-02750-t001:** Clinical data of patients with ulcerative colitis and healthy controls.

Variable	Patients with Ulcerative Colitis (n = 40)	Healthy Controls (n = 25)	*p*-Value	Statistical Significance
Age (years), mean ± S.D.	48.5 ± 17.3	43.8 ± 10.7	0.237	-
Female/male (%)	19/21 (47.5/52.5)	17/8 (68.0/32.0)	0.129	-
Weight (kg), mean ± S.D.	66.10 ± 14.41	66.10 ± 11.28	0.445	-
Height (cm), mean ± S.D.	164.01 ± 7.19	165.31 ± 9.65	0.593	-
Body mass index, mean ± S.D.	24.38 ± 3.81	23.30 ± 3.86	0.305	-
White blood cell, mean ± S.D.	6.34 ± 2.01	6.76 ± 1.71	0.398	-
Red blood cell, mean ± S.D.	4.48 ± 0.51	4.54 ± 0.49	0.642	-
Hemoglobin, mean ± S.D.	13.76 ± 2.00	13.84 ± 1.76	0.868	-
Hematocrit, mean ± S.D.	39.15 ± 9.50	43.16 ± 5.57	0.061	-
Platelet, mean ± S.D.	251.79 ± 56.39	294.08 ± 56.84	0.005	**
Segmented neutrophil	58.67 ± 9.51	-^(6)^	-	-
Lymphocyte	31.49 ± 8.37	-	-	-
Monocyte	8.10 ± 5.16	-	-	-
Eosinophil	2.17 ± 1.62	-	-	-
Basophil	0.68 ± 0.45	-	-	-
Calcium, mean ± S.D.	16.49 ± 44.53	9.52 ± 0.37	0.438	-
Phosphorus, mean ± S.D.	3.67 ± 1.29	4.34 ± 2.12	0.165	-
Glucose, mean ± S.D.	101.88 ± 39.66	94.04 ± 11.82	0.264	-
Creatinine, mean ± S.D.	3.30 ± 15.33	0.72 ± 0.19	0.404	-
eGFR ^(1)^, mean ± S.D.	101.46 ± 13.23	104.96 ± 26.74	0.549	-
BUN ^(2)^, mean ± S.D.	15.76 ± 12.54	11.87 ± 3.00	0.134	-
Uric acid, mean ± S.D.	5.11 ± 1.64	4.83 ± 1.24	0.476	-
Cholesterol, mean ± S.D.	187.43 ± 37.77	210.16 ± 49.21	0.044	*
Protein, mean ± S.D.	10.59 ± 22.72	7.45 ± 0.47	0.493	-
Albumin, mean ± S.D.	4.48 ± 0.61	4.81 ± 0.31	0.017	*
AST ^(3)^, mean ± S.D.	21.77 ± 9.81	22.48 ± 5.72	0.746	-
ALT ^(4)^, mean ± S.D.	21.22 ± 11.68	19.60 ± 10.05	0.574	-
ALP ^(5)^, mean ± S.D.	69.78 ± 18.93	63.80 ± 15.68	0.197	-
Total bilirubin, mean ± S.D.	0.56 ± 0.22	1.01 ± 1.94	0.256	-

^(1)^ eGFR: estimated glomerular filtration rate. ^(2)^ BUN: blood urea nitrogen. ^(3)^ AST: aspartate aminotransferase. ^(4)^ ALT: alanine aminotransferase. ^(5)^ ALP: alkaline phosphatase. ^(6)^ -: not-tested. The *p*-value was calculated using the independent sample *t*-test; * *p* < 0.05, ** *p* < 0.01.

## Data Availability

The data presented in this study are available upon request from the corresponding author.

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
