# Peer review of "Comparative Study of Intestinal Microbiome in Patients with Ulcerative Colitis and Healthy Controls in Korea"

_microorganisms, 2023, doi:10.3390/microorganisms11112750_

Round 1
Reviewer 1 Report
Comments and Suggestions for Authors
Thank you for the opportunity to review this paper. Most of the content are correct, although some parts of the manuscript could be improved for content and clarity.
Specific comments:
1. As per scientific nomenclature, you need to italicize family, genus, species, and variety or subspecies. Kingdom, phylum, class, order, and suborder begin with a capital letter but are not italicized.
2. "Ulcerative colitis (UC) is the contemporary challenging medical condition ..." - odd expression, consider specifying why it is challenging (e.g., rising incidence, diagnostic challenges, management difficulties).
3. Please change "rectal urgency" to "fecal urgency".
4. "We recruited 40 patients with UC and 25 non-patients which had no medicine nor treatment" - when you say non-patients, do you mean they do NOT have UC or could they have UC but just do not have medicine or treatment for it? Please clarify. Specifying the characteristics of ‘non-patients’ or using a term like ‘healthy controls’ might provide better clarity.
5. Were the UC patients confirmed to be clinically and biochemically quiescent? Did you take serum acute-phase reactants and fecal calprotectin levels?
6. Did the authors perform microbiological confirmation?
7. While the different abundances of Firmicutes and Bacteroidota are noted, the authors did not go far enough to discuss why this might be relevant or interesting in the context of UC.
8. Similarly, while there is mention of the potential use of the findings, it may be helpful to briefly note why or how these microbiome differences might be mechanistically relevant in UC or elaborate briefly on how it could inform future treatments.
9. It is relevant to mention that in a recent systematic review, the authors found that studies have implicated Bacteroides, Faecalibacterium prausnitzii, Ruminococcus spp., and Bifidobacteria in irritable bowel syndrome (which is also associated with mucosal inflammation) and treatment response but there was significant heterogeneity among the studies and no uniform characteristics of IBS-related gut microbiota has been identified (citation: pubmed.ncbi.nlm.nih.gov/37110143).
10. I would appreciate it if the authors could give an outlook for future directions based on their summarized results here. For instance, do we need more time-series measurements or larger sample sizes?
11. Please correct the duplicate numbering in the references.
Comments on the Quality of English LanguageModerate edits needed.
Author Response
<Response to Reviewer 1>
Comment 1: As per scientific nomenclature, you need to italicize family, genus, species, and variety or subspecies. Kingdom, phylum, class, order, and suborder begin with a capital letter but are not italicized.
Response: As your suggestion, we have revised the higher taxonomic categories (order, class, phylum, kingdom) with capital letters and without italicization throughout the manuscript.
Comment 2: "Ulcerative colitis (UC) is the contemporary challenging medical condition ..." - odd expression, consider specifying why it is challenging (e.g., rising incidence, diagnostic challenges, management difficulties).
Response: As your suggestion, we modified line 19-21.
Line 19-21:
Ulcerative colitis (UC) is the contemporary challenging medical condition; however, its etiology remains unclear.
⇒
Ulcerative colitis (UC) poses a contemporary medical challenge, with its exact cause still eluding researchers. This is due to various factors, such as the rising incidence, diagnostic complexities, and difficulties associated with its management.
Comment 3: Please change "rectal urgency" to "fecal urgency".
Response: As your suggestion, we have changed “rectal urgency” to “fecal urgency”.
Line 42:
Characteristic symptoms of UC include bloody diarrhea, rectal urgency, and tenesmus
⇒
Characteristic symptoms of UC include bloody diarrhea, fecal urgency, and tenesmus
Comment 4: "We recruited 40 patients with UC and 25 non-patients which had no medicine nor treatment" - when you say non-patients, do you mean they do NOT have UC or could they have UC but just do not have medicine or treatment for it? Please clarify. Specifying the characteristics of ‘non-patients’ or using a term like ‘healthy controls’ might provide better clarity.
Response: As your suggestion, we have changed “non-patients” to “healthy-controls” throughout the manuscript.
Comment 5: Were the UC patients confirmed to be clinically and biochemically quiescent? Did you take serum acute-phase reactants and fecal calprotectin levels?
Response: All participant were diagnosed with UC at Chungbuk National University, and were received treatment. The diagnosis of UC was based on clinical, endoscopic, and histological information by a experienced gastroenterologist doctor. Some samples, but not all samples from patients were analyzed acute-phase reactants and/or fecal calprotectin level.
Comment 6: Did the authors perform microbiological confirmation?
Response: Thank you for this suggestion. It would have been interesting to explore this aspect. Unfortunately, we didn’t perform microbiological confirmation including bacterial culture, and identification. All fecal samples were stored frozen for long-time, we couldn’t perform microbiological confirmation for these. Next time, we’ll perform microbiological confirmation according to your suggestion.
Comment 7: While the different abundances of Firmicutes and Bacteroidota are noted, the authors did not go far enough to discuss why this might be relevant or interesting in the context of UC.
Response: According to your suggestion, we added discussion for Firmicutes and Bacteroidota on line 322-332 and line 337-342
Line 322-332:
Our study demonstrated that Firmicutes was the most prevalent phylum in both groups, and patients with UC had more prevalence of Firmicutes compared to healthy-controls (patients with UC: 51.12%, and healthy-controls: 46.90%, respectively). The levels of bile acids might be altered in patients with UC [1, 8, 20]. Changes in the composition of fecal bile acids might be involved in modulating inflammatory responses [6, 8, 20]. The primary bile acids are transformed into secondary bile acids by the intestinal microbiota through various reactions, such as deconjugation, dehydroxylation, esterification, and desulfatation [6, 8, 9]. Dehydroxylation is facilitated by the Firmicutes [3, 8, 9]. Therefore, an imbalance in gut microbiota could disrupt the process of bile acids, potentially leading to elevated levels of secondary bile acids in patients with UC, and contribute to the out-break of UC.
Line 337-342:
Bacteroidetes, along with other beneficial bacteria, play a role in maintaining gut homeostasis by contributing to processes such as digestion, nutrient absorption, and immune regulation [36]. Changes in the composition of the gut microbiota, including a reduction in beneficial bacteria like Bacteroidetes, can disrupt these processes and potentially contribute to inflammation and other symptoms associated with ulcerative colitis [36].
Comment 8: Similarly, while there is mention of the potential use of the findings, it may be helpful to briefly note why or how these microbiome differences might be mechanistically relevant in UC or elaborate briefly on how it could inform future treatments.
Response: According to your suggestion, we added potential use of the findings on line 433-437 and line 451-453.
Line 433-437:
In this study, differences of gut microbiota were observed in the abundance at the phylum and genus levels, as well as in the intestinal microflora structure between patients with UC and healthy controls. Additionally, patients with UC exhibited lower community richness compared to healthy individuals. Our findings will provide a foundation for re-searching treatments focused on modifying the intestinal microbiome of patients with UC.
Line 451-453:
In future research, it would be beneficial to investigate bacterial species, their functions, and their correlation with host genetic factors. This approach could enhance our under-standing of the etiology of UC.
Comment 9: It is relevant to mention that in a recent systematic review, the authors found that studies have implicated Bacteroides, Faecalibacterium prausnitzii, Ruminococcus spp., and Bifidobacteria in irritable bowel syndrome (which is also associated with mucosal inflammation) and treatment response but there was significant heterogeneity among the studies and no uniform characteristics of IBS-related gut microbiota has been identified (citation: pubmed.ncbi.nlm.nih.gov/37110143).
Response: As your suggestion, we added Reference 27, and line 104-106.
Ref. 27:
27. Ng Q. X.; Yau C. E.; Yaow C. Y. L.; Chong R. I. H.; Chong N. Z-Y.; Teoh S. E.; Lim Y. L.; Soh A. Y. S.; Ng W. K.; Thumboo J. What Has Longitudinal ‘Omics’ Studies Taught Us about Irritable Bowel Syndrome? A Systematic Review. Metabolites 2023, 13, 484.
Line 104-106:
In a recent systematic review, significant heterogeneity was observed among studies on gut microbiota, and no consistent characteristics of irritable bowel syndrome-related gut microbiota have been identified [27]
Comment 10: I would appreciate it if the authors could give an outlook for future directions based on their summarized results here. For instance, do we need more time-series measurements or larger sample sizes?
Response: As outlined in the manuscript, the study's limitations involve a small sample size and the methodology employed. We analyzed gut microbiota at genus level however, for better understanding the relationship between ulcerative colitis and gut microbiota, analysis at the species level would be better. Also, larger sample size could have more statistical power. Time-series measurements could also give us insight for etiology of ulcerative colitis however,
According to your suggestion, we modified line 439-453.
Line 439-453:
Limitations of this study include a small sample size and the employed methodology. In this study, we found that there was a tendency to differ in gut microbiota between patients with UC and healthy-controls however, statistical significance was not clearly demonstrated. In the future, conducting studies with a larger sample size could provide us with greater statistical power. Also, because UC had a various contributory factors including disturbances in gut microbiota stability, immune system imbalances, genetic pre-disposition, and unhealthy lifestyle choices, it is important to ensure that these factors are controlled in the sample. In this study, the microbial community characterization was conducted using the MiSeq Illumina platform, based on the common V3-V4 fragments of the 16S rRNA gene. Nonetheless, it has been shown that the degree of variation in different microbial taxa may be influenced by the regions (such as V1-V9) analyzed in the 16S rRNA gene [30]. Using primers that target specific groups of bacteria may lead to limitations in the detection and targeting of other microbial groups, thus potentially limiting the comparability of results between studies. In future research, it would be beneficial to investigate bacterial species, their functions, and their correlation with host genetic factors. This approach could enhance our understanding of the etiology of UC.
Reviewer 2 Report
Comments and Suggestions for Authors
Reviewer comments and suggestions
The authors in this study compared the intestinal microbiome of patients with Ulcerative colitis (UC) to that of non-patients to determine the qualitative and quantitative changes associated with UC that occur in the intestinal microbiota.The intestinal bacterial abundance in 40 Korean patients with UC and 25 non-patients was assayed using 16S rRNA gene-based analysis via next-generation sequencing.
There were five major phyla in both groups: Firmicutes, Bacteroidota, Proteobacteria, Actinobacteriota, and Desulfobacteriota. The authors found that the intestinal microbiome of patients with UC was less diverse, and bacterial abundance was different between patients with UC and non-UC individuals. The Firmicutes were more prevalent in patients with UC (51.12%) compared to that of non-patients (46.90%). Otherwise, the Bacteriodota was more prevalent in non-patients (40.34%) compared to patients with UC (37.04%). With the help of other analysis, the authors stated new treatment options and lay the groundwork for future research on UC.
Overall, the manuscript was well written. However, a few major concerns or comments needed to be explained or modified.
- Line 44-45 It would be nice if the authors could elaborate about the cause of UC for the common reader of your manuscript
- Line 73-74 Please explain it well
- Line 90-91 These cited studies are not well described
- Line 93-94 What would be the possible reason for this
- In the material method section they did not discuss the tools and as well statistics part in the manuscript, please add up, it’s important
- Line 175-177 So there was no difference in patients and nonpatients gp in biochemical analysis, please check with other published studies
- Comments for discussion first paragraph The authors have to point out the novelty of this study in the first paragraph
- Please add up a table or figure in the discussion text for easy following your manuscript
- Still, I could not get the conclusion present in the manuscript, please modify the conclusion part
Author Response
<Response to Reviewer 2>
Global comment
he authors in this study compared the intestinal microbiome of patients with Ulcerative colitis (UC) to that of non-patients to determine the qualitative and quantitative changes associated with UC that occur in the intestinal microbiota.The intestinal bacterial abundance in 40 Korean patients with UC and 25 non-patients was assayed using 16S rRNA gene-based analysis via next-generation sequencing.
There were five major phyla in both groups: Firmicutes, Bacteroidota, Proteobacteria, Actinobacteriota, and Desulfobacteriota. The authors found that the intestinal microbiome of patients with UC was less diverse, and bacterial abundance was different between patients with UC and non-UC individuals. The Firmicutes were more prevalent in patients with UC (51.12%) compared to that of non-patients (46.90%). Otherwise, the Bacteriodota was more prevalent in non-patients (40.34%) compared to patients with UC (37.04%). With the help of other analysis, the authors stated new treatment options and lay the groundwork for future research on UC.
Overall, the manuscript was well written. However, a few major concerns or comments needed to be explained or modified.
Comment 1: Line 44-45 It would be nice if the authors could elaborate about the cause of UC for the common reader of your manuscript
Response: Thank you for this suggestion. However, in this section, we wanted to inform for the common readers that cause of UC are unknown. So, we think it would be better that this sentence would not be modified.
Comment 2: Line 73-74 Please explain it well
Response: As your suggestion, we revised line 77-81.
Line 77-81: Hörmannsperger et al. reported that some immunodeficient mice kept under conventional condition developed chronic colitis spontaneously; however, if mice were kept under germ-free conditions or treated long-term antibiotic therapy, the spontaneous development of colitis would be prevented [19].
Comment 3: Line 90-91 These cited studies are not well described
Response: According to your suggestion, line 93-102 were revised and added.
Line 93-102:
Furthermore, numerous studies conducted by other researchers have indicated that dysbiosis associated with UC often involves a decrease in beneficial commensal bacteria, particularly those within the Firmicutes and Bacteroides phyla, coupled with an increase in pathogenic species from the Enterobacteriaceae family [23–26]. C. Casen et al. reported 80% of IBD patients had dysbiosis characterizing reduced Firmicutes [23],and, Alan W Walker et al. reported that microbial diversity was reduced in IBD patients, and Firmicutes were reduced [24]. Zhang M et al. [25] and Zhang S-L et al. [26] demonstrated that Bacteroides in the gut microbiota of patients with UC had significant changes in amount and had roles in the onset and progression of IBD.
Comment 4: Line 93-94 What would be the possible reason for this
Response: According to your suggestion, we added the possible reason for ‘no substantial differences between patients with UC and healthy indiciduals” on line 106-110.
Line 106-110:
Willing et al. reported that significant differences between individuals with UC and healthy individuals were not observed, and assumed that this lack of distinction was at-tributed to the fact that environmental exposure during early childhood exerted a more profound and enduring influence on the gut microbiota compared to disease status [28]
Comment 5: In the material method section they did not discuss the tools and as well statistics part in the manuscript, please add up, it’s important
Response: According to your suggestion, we added materials & method section on line 143-198
Line 143-171
DNA extraction
Fresh fecal samples from participants were collected in a sterile container, and then stored in the deep-freezer at -80 oC. Bacterial DNA was extracted from 200 mg of fecal sample collected from each individual using the Maxwell RSC Instrument (Promega, Madison, WI, USA) and the Maxwell RSC Fecal Microbiome DNA Kit AS1700 (Promega), according to the manufacturer’s protocol. After the extraction, the quality of DNA were assayed using a Qubit dsDNA HS assay kit (Life Technologies, Gent, Belgium) in a Qubit fluorometer (Thermo Fisher Scientific, USA) according to the manufacturer’s protocol.
16S rRNA gene sequencing
After quantification using a Qubit dsDNA HS assay kit (Life Technologies, Gent, Belgium) in a Qubit fluorometer (Thermo Fisher Scientific, USA), 65 16S V3-V4 amplicon li-braries (40 libraries for patients with UC and 25 libraries for healthy-controls) were pre-pared according to the Illumina metagenomic sequencing library construction workflow (Part # 15,044,223 Rev. B, Illumina, San Diego, CA, USA). Briefly, sequences of the region V3-V4 of the 16S rRNA bacterial gene were amplified using the KAPA HiFi HotStart ReadyMix PCR Kit (KAPA Biosystems, Wilmington, MA, USA) and the following primers (16S rRNA gene-specific sequences are underlined): for-ward;5’-TCGTCGGCAGCGTCAGATGTGTATAAGAGACAGCCTACGGGNGGCWGCAG-3’ and re-verse;5’-GTCTCGTGGGCTCGGAGATGTGTATAAGAGACAGGACTACHVGGGTATCTAATCC-3’. Subsequently, the second PCR was carried out to attach the index adapters for sample multiplexing using the Nextera XT Index Kit (Illumina, San Diego, CA, USA). The concentrations of each DNA libraries were quantified by a Qubit fluorometer (Thermo Fisher Scientific, USA). The DNA libraries were pooled at concentrations of 4 nM and de-natured with 0.2 N NaOH. The final concentration of the library was 7 pM, and the phiX control library was added at 30% (v/v) at the same concentration as that of the library. The sequencing of the libraries was carried out on the MiSeq platform according to the manu-facturer’s instructions, using the MiSeq Reagent Kit v3 (600 cycles) (Illumina).
Line 176-180:
The forward and reverse reads were merged and matched to respective samples using their assigned indices. Subsequently, raw reads were trimmed using QIIME. Quality fil-tering was applied to the combined sequences, discarding sequences that did not meet the following criteria: sequence length < 20 nucleotides or > 300 nucleotides.
Line 181-183:
Prior to computing alpha- and beta-diversity statistics, the sequences were rarefied. The sufficiency of the sample size and estimation of species richness were determined using a rarefaction curve.
Line 185-189:
Alpha diversity statistics were calculated in QIIME 2. Beta-diversity was assessed through the utilization of Bray-Curtis distances, and subsequently, a principal coordinate analysis was conducted. The differences in microbial diversity among samples (beta diversity) were measured using Bray-Curtis distances and represented visually through multidi-mensional scaling plots.
Comment 6: Line 175-177 So there was no difference in patients and nonpatients gp in biochemical analysis, please check with other published studies
Response: I don’t know exact means of “gp” that you said however, if you want to say “gpt” , I know that gpt is not associated with ulcerative colitis generally. If you mean “general protein”, patients with UC showed higher total protein levels compared to healthy-controls however, there was no significant difference. This is presumed to be due to a large difference between individuals in blood chemistry tests and a small number of samples.
Comment 7: Comments for discussion first paragraph The authors have to point out the novelty of this study in the first paragraph
Response: As your suggestion, we revised the first paragraph of discussion.
Line 295-299:
In this study, we investigated the microbiological composition of the intestines of patients with UC in Korea and compared the data with those of a healthy-control group. To the best of our knowledge, this study represents the first analysis of the gut microbiota in Korean patients with UC and healthy controls, incorporating clinical data such as complete blood cell counts and biochemical analysis.
Comment 8: Please add up a table or figure in the discussion text for easy following your manuscript
Response: Thank you for this suggestion. However, we believe readers will fully understand the table and figures in the results section, and adding a new figure in the discussion section might cause confusion.
Comment 9: Still, I could not get the conclusion present in the manuscript, please modify the conclusion part
Response: As your suggestion, we modified conclusion part on line 433-453.
Line 433-453:
In this study, differences of gut microbiota were observed in the abundance at the phylum and genus levels, as well as in the intestinal microflora structure between patients with UC and healthy controls. Additionally, patients with UC exhibited lower community richness compared to healthy individuals. Our findings will provide a foundation for re-searching treatments focused on modifying the intestinal microbiome of patients with UC.
Limitations of this study include a small sample size and the employed methodology. In this study, we found that there was a tendency to differ in gut microbiota between patients with UC and healthy-controls however, statistical significance was not clearly demonstrated. In the future, conducting studies with a larger sample size could provide us with greater statistical power. Also, because UC had a various contributory factors in-cluding disturbances in gut microbiota stability, immune system imbalances, genetic pre-disposition, and unhealthy lifestyle choices, it is important to ensure that these factors are controlled in the sample. In this study, the microbial community characterization was conducted using the MiSeq Illumina platform, based on the common V3-V4 fragments of the 16S rRNA gene. Nonetheless, it has been shown that the degree of variation in different microbial taxa may be influenced by the regions (such as V1-V9) analyzed in the 16S rRNA gene [30]. Using primers that target specific groups of bacteria may lead to limitations in the detection and targeting of other microbial groups, thus potentially limiting the comparability of results between studies. In future research, it would be beneficial to investigate bacterial species, their functions, and their correlation with host genetic factors. This approach could enhance our understanding of the etiology of UC.

Round 2
Reviewer 1 Report
Comments and Suggestions for Authors
Thank you for the revisions.
Specific comments:
1. "Bacteriodota" - spelling error.
2. Lack of microbiological confirmation is also a limitation that should be mentioned.
3. Please provide information on the ethical approval for the study, e.g. "This study was approved by the XXX ethics Committee of XXX Hospital/University (registration number, date)".
Comments on the Quality of English LanguageMinor edits.
Author Response
<Response to Reviewer 1>
Comment 1: “Bacteroidota” – spelling error
Response: As your suggestion, we revised spelling error “Bacteriodota” to “Bacteroidtoa”
In addition to the above comment, all spelling and grammatical errors have been checked again.
Line 30: Bacteriodota -> Bacteroidota
Comment 2: Lack of microbiological confirmation is also a limitation that should be mentioned.
Response: As your suggestion, we added limitations of study on line 448-451.
Line 448-451:
If microbial confirmation including culture and identification had been performed, it would have been more helpful in understanding the gut microbiota of patients with UC and healthy controls. However, in this study, microbial confirmation was not conducted.
Comment 3: Please provide information on the ethical approval for the study, e.g. "This study was approved by the XXX ethics Committee of XXX Hospital/University (registration number, date)".
Response: We’ve already provided information on the ehtical approval for the study on line 126-128.
According to your comment, we moved this sentence to line 122-124.
Line 122-124:
Ethics statement
This study was approved by the Institutional Review Board of Chungbuk National University (Registration number: CBNUH CTC-21-04; Effective date: November 11, 2021).
